# Prospective Study on the Excretion of Mucous Stools and its Association with Age, Gender, and Feces Output in Captive Giant Pandas

**DOI:** 10.3390/ani9050264

**Published:** 2019-05-22

**Authors:** Zixiang Li, Xuefeng Liu, Juan Zhao, Yanhui Liu, Haihong Xu, Changqing Li, Tao Ma, Bo Wang, Yanping Lu, Barbara Padalino, Dingzhen Liu

**Affiliations:** 1Ministry of Education, Key Laboratory of Biodiversity Sciences and Ecological Engineering, College of Life Sciences, Beijing Normal University, Beijing 100875, China; 201721200064@mail.bnu.edu.cn; 2Beijing Key Laboratory of Captive Wildlife Technologies in Beijing Zoo, Beijing 100044, China; lxf9722@163.com (X.L.); zhaojuanfly@163.com (J.Z.); huihuia6@sina.com (Y.L.); 13911619616@163.com (H.X.); hupi001@sina.cn (C.L.); taoshengyijiu034@Sina.com (T.M.); w35143245@sina.com (B.W.); 3Department of Veterinary Medicine, University of Bari, 70010 Bari, Italy; barbara.padalino@uniba.it; 4Jockey Club College of Veterinary Medicine and Life Sciences, City University of Hong Kong, Kowloon, Hong Kong, China

**Keywords:** giant panda, mucous stool, bamboo preference, feces output

## Abstract

**Simple Summary:**

Captive pandas may excrete mucous stools, which are often accompanied by discomfort and decreased activity. Thus, the excretion of mucous stools is a welfare concern for pandas kept in zoos. In this study, we documented the frequency of mucous excretions and the daily feces weights of 18 giant pandas (*Ailuropoda melanoleuca*) in the Beijing Zoo from April 2003 to June 2017. We also explored possible associations between the excretion of mucous stools and the pandas’ gender, age, and feces output. During the study, 900 cases of mucous excretion occurred over 32,856 observation days. The frequency of mucous excretion was negatively correlated with the feces output and positively correlated with the pandas’ age. No correlation with gender was found. Moreover, the mean frequency of mucus occurrence showed monthly changes, with one significant peak in October, and the time series analyses showed that the time (month) change imposed the strongest negative effect on fecal output, while the biggest effects on fecal output and mucus excretion both occurred in August (seasonal factors were −2.261 and 0.0126, respectively). Our findings not only add to the literature concerning pandas but are also helpful for enhancing their welfare. Further studies addressing the underlying mechanisms for the occurrence of mucus in captive pandas are needed.

**Abstract:**

The giant panda (*Ailuropoda melanoleuca*) has evolved a large number of mucous glands in the intestinal lining to adapt to the digestion of high-fiber foods. However, in captive pandas, excessive mucus might form a mass and then be eliminated, which is often accompanied by discomfort and decreased activity. This event is called ‘mucous excretion’. The causes of mucus excretions in captive pandas, however, remain unknown. The aims of this study were to document the occurrence of mucus excretion and to investigate its possible associations with pandas’ age, gender, and feces output. Eighteen giant pandas were studied at the Beijing Zoo from April 2003 to June 2017, and a total of 900 occurrences of mucous excretion and 32,856 daily defecation outputs in weight were recorded. The likelihood of mucous excretion occurrence decreased by 11.34% for each 1 kg of fecal output (Z = −4.12, *p* < 0.0001), while it increased by 5.89% per year of age (Z = 4.02, *p* < 0.0001). However, individual differences in gender had no significant effect on the mucous occurrence (Z = −0.75, *p* = 0.4508). A monthly change in mucus occurrence was also found. The mean frequency of mucus occurrence was significantly higher in October. In August, time (month) change showed the biggest negative influence on feces output but the biggest positive influence on mucus excretion (seasonal factors were −2.261 and 0.0126, respectively). Our results documented the occurrence of mucous excretions and confirmed their possible associations with the pandas’ age and fecal output based on a 15-year prospective study. This study not only adds to our knowledge of panda physiology but also suggests the need for further studies examining the causes of the excretion of mucous stools in captive pandas. Reducing the incidence of mucous excretion would promote ex situ conservation and enhance panda welfare.

## 1. Introduction

Zoos, breeding centers, and sanctuaries play a major role in conserving wild animals, particularly for some endangered species. However, many captive animals may show physio-pathological responses to the diets provided by humans. For example, the captive spectacled bear (*Tremarctos ornatus*) will excrete loose stools if grass or other fiber sources are not available [1]. Similarly, the captive giant pandas (*Ailuropoda melanoleuca*) and red pandas (*Ailurus fulgens*) may defecate mucous stools when the welfare principle of good feeding is not met [2,3]. Several previous studies have shown that diet transition may stimulate the excretion of mucous stools [4,5]. However, while the seasonal diet changes from leaves to culm or vice versa occur periodically in wild pandas, the occurrence of mucous stool excretion has never been documented in the wild. Consequently, it was proposed that mucous stool excretion might be related to captive husbandry and/or management [6]. Nickley et al. conducted a study in two pandas at the San Diego Zoo and proposed that low bamboo intake could account for the excretion of mucous stools [6]. Similar findings were reported in a three-year biomedical survey of 61 captive giant pandas in a variety of Chinese institutions [7]. However, a long-term prospective study on the occurrence of mucous stools in giant pandas has never been carried out. 

The giant panda was an endangered species before 2016 and is now listed as a vulnerable animal species in China, because the wild population has shown an increase [8]. In November 2018, both the captive and wild populations showed an increase, with a total of 548 and 1864 subjects, respectively [9]. However, the natural mating rate is still less than 40% on average (range 20–80%, [10]). Mucus excretion (Figure 1a) is a common and frequent phenomenon for pandas in captivity. Whenever this happens, the panda becomes inactive and in a bad mood, loses its appetite, crouches for long periods, and emits groan vocalizations (Dingzhen Liu personal observations, Figure 1b). This phenomenon usually lasts for one to three days. If mucous events coincide with the single estrus that a female has each year on a regular basis, this could lead to negative effects on captive reproduction [11]. For successful ex situ conservation, it is therefore essential to reduce the incidence of mucus stool excretion in captive pandas. 

Unlike other bears, the giant panda is a specialized bamboo feeder [12]. It has a carnivorous digestive system and lacks the genes coding for the enzymes required to digest cellulose [13]. However, the giant panda has developed a special adaption to its bamboo diet with the help of gut microbes [14]. Meanwhile, they also have a special gastrointestinal gland and specialized cells, called ‘Jejunal goblet cells’, on the inner surface of the intestinal tract. The latter protect the inner surface of the intestine from being hurt by the pieces of bamboo branches by excreting mucus [5]. For wild individuals, bamboo accounts for 99% of their daily diet. It was reported that an adult female (“Zhenzhen”) ate 38.3 kg bamboo shoots on average per day [12], and a wild adult panda ate between 43.6 and 57.0 kg bamboo shoots daily at the Foping Natural Reserve [15,16]. A previous study showed that the more bamboo pandas eat, the more feces they will produce [3]. Thus, the mucus can be excreted gradually by coating a small amount on the surface of the panda droppings. Mucus excretion has never been reported in wild pandas. In captive pandas, the daily ratio of fresh bamboo is greatly reduced (56% and 75% of their total dry matter ratio [6]; the intake of bamboo in a captive adult panda is 3.4–9.4 kg/day, with an average of 6.0 kg/day [17]), and mucus excretion is a common health problem. 

The aim of this 15-year prospective study was to document the occurrence of mucus excretion, the individual daily fecal output, and the monthly variations in pandas kept in zoos. Possible associations between the frequency of the excretion of mucous stools and gender, age, and individual daily fecal output were also explored. The results will increase our knowledge on the occurrence of mucous stools and will benefit the husbandry, management, and ex situ conservation of this vegetarian carnivore species.

## 2. Materials and Methods

### 2.1. Subjects and Management

The data of giant pandas (*n* = 18), aged from 1 to 31 years old, at the Beijing Zoo, from April 2003 to June 2017 were included in this study. The gender, age, and studbook numbers for these pandas are shown in Appendix A. The pandas were individually kept in enclosures that included an exhibition indoor pen (5.0 × 10.0 m) and an outdoor yard (10.0 × 50.0 m) with an artificial stone hill. There was a small pond (1.0 m in diameter) as the water source. 

Each subject was fed daily with steamed bread (0.8–1.2 kg/day), fresh carrots (0.5 kg/day), and one apple. Fresh bamboo was collected in Henan Province, China, and transported to Beijing by a land route at room temperature on the previous day and was provided ad libitum. The daily average weight of the bamboo provided to each panda was approximately 25.0 ± 5.0 kg. However, from our preliminary observation in the Beijing Zoo and the China Conservation and Research Center for the Giant Panda (CCRCGP), the studied captive giant pandas only ate 40–50% of the bamboo provided daily. Since January 2016, bamboo shoots (2.0–4.0 kg/day) were also provided to each panda throughout the year at the Beijing Zoo. 

During our study, two female pandas “Jini” and “Niuniu” were temporarily translocated from the Beijing Zoo to the CCRCGP for collaborative annual breeding in the spring. In addition, six individuals died of natural causes during the study. Thus, the period of data collection for each individual ranged from 10 to 171 months (64.8 months on average). Detailed information about the translocation history, death date, and data collection duration for each individual is shown in Appendix A.

### 2.2. Experimental Design

#### 2.2.1. Fecal and Mucus Stool Samples Collection

All fecal droppings excreted from 8:00 am on a previous day to 8:00 am on observation day were collected, weighed using an electric scale (TGT-500A, Beijing Xuanwu Weighing Apparatus Factory, Beijing, China) (precision 0.1 kg), and recorded for each panda while the zookeeper conducted the daily routine of cleaning. Meanwhile, the keeper also observed the animal’s behavior, in particular documenting whether the panda was inactive (staying in one place for a period longer than 1 h without moving) and emitting groaning vocalizations. The zookeeper further examined whether the animal was excreting or had excreted mucous stools (Figure 1a) during the same time window as the fecal sample collection. The observation day was marked as “Yes” if the animal excreted a mucous stool and was otherwise marked as “No” in the notebook.

#### 2.2.2. Statistics

We first calculated the daily feces output and frequency of mucus excretion for each panda, and then used that data in the subsequent correlation analysis. We used the generalized estimating equations (GEE) model of Proc Genmod with SAS (SAS Institute Inc., Cary, NC, USA), which allowed us to adjust for the repeated measurements as clusters to perform the analysis. In this model, we used the occurrence of mucus (1 for “Yes” and 0 for “No”) as the dependent variable and the daily fecal amount as the independent variable. To examine the potential effects of each animal’s individual variation on their mucus excretions, we used age and gender as the estimates of individual differences in the GEE model analysis. In addition, we examined the seasonal changes in the mean frequency of mucous occurrences and fecal amount. We used an iterative process to identify elevated peaks in the monthly frequency of mucous occurrences using the method of Brown et al. [18]. In brief, we first calculated the means and standard deviation of the mean (using a boxplot) in each month for all individuals throughout all the years in our study. Next, we calculated the average and standard deviation of those above 12 monthly mean data, and monthly values falling beyond two SD from the average (i.e., the peaks) were considered significantly different and were excluded from further calculations. This process was repeated until no new significant peaks were identified. To explore the monthly influence on the frequency of mucus occurrence and feces output, we also performed a time series analysis of the months with aggregated years and individuals. To run this analysis, we defined ‘month’ as the time factor by using the additive model. The monthly factors in this analysis represent the time effects (negative or positive) on the change in fecal output or mucous excretion after eliminating the trend and random effects. The absolute value of the monthly factor indicates the degree of the time effects on the fecal output and mucus occurrence. Moreover, we calculated the average feces output and mucous stool excretion occurrence per month for each panda and drew contour maps for each panda to show the changes in the mean monthly feces amounts and mean monthly frequencies for the mucus occurrences of each panda within each month and year. 

GEE model was run with SAS (V9.4); all other statistical analyses were run with SPSS (V22.0) (IBM, Armonk, NY, USA). All of the tests were two-tailed. The alpha level was set at 0.05. 

## 3. Results

We recorded 900 mucus excretion events over 32,856 observation days, with an average daily feces output of 7.28 ± 0.02 kg (Median: 5.0; Minimum: 0.00; Maximum 34.00). The contour maps of the feces and mucus for each panda are shown in Appendix A. The feces output and mucus occurrence showed monthly changes, even though there were strong individual differences. The peaks for feces output occurred mainly in two periods, i.e., from February to June and October to December (Appendix A). The highest mucous occurrence frequencies showed high variances as the months changed from January to December (Appendix A), and 6/14 of pandas showed a trend of mucous excretion peaks in February–March and October–November.

The results of the GEE analysis demonstrated that this model algorithm converged, and the results were reliable for further analysis (Table 1). Further analyses showed that gender was not associated with the occurrence of mucous stools (*p* = 0.5257). However, the daily feces excretion had a significant and negative influence on the occurrence of mucous stool excretion (Z = −4.12, *p* < 0.0001), while age showed a significant and positive effect on the mucous stool excretion (Z = −4.02; *p* < 0.0001) in the studied pandas (Table 2). Consequently, when the daily amount of the feces was lower, there was a higher likelihood of mucous excretion in those pandas. In particular, the probability of excreting mucous stools decreased by 11.34% for each 1 kg increase in feces. We found that older pandas were more likely to excrete mucous stools; in particular, the probability of excreting mucous stools increased by 5.89% for each year of age (Table 2). 

The boxplots and peak identify analyses of the mean frequency of mucus occurrence per month showed only one significant peak in October (0.05 ± 0.02 times/individual/year), and no peaks were found in the mean monthly feces output (Figure 2). The time series analysis results showed the seasonal factor in each month (Table 3). The influence of the seasonal change in August was significantly greater than any other month in both mucus occurrence (0.0126) and feces output (−2.261). We also found that 15/17 individuals were defecating significantly less in August than the annual median for the whole population at the Beijing Zoo throughout the 15 years of the study (data not shown).

## 4. Discussion

This 15-year prospective study documented the occurrence of mucous stools and the daily feces output of 18 captive pandas kept at the Beijing Zoo. In our study, the occurrence of this phenomenon was slightly higher (2.74%) than that reported in a previous study conducted at the Chengdu Panda Base (2.16%) [11]. Our study also investigated the possible association between the excretion of mucous stools and the pandas’ age, gender, and feces output. Our results suggest that the occurrence of mucous stools in captive giant pandas was negatively associated with the daily feces excretion amount and was positively and significantly associated with the pandas’ age. The feces output and mucous stool occurrence of each of the individual pandas showed seasonal changes with the month in contour maps (Appendix A). Meanwhile, in the studied pandas, the mean frequencies of occurrence of mucous stools were the highest in October, while the time series analyses showed that time (month) change imposed the greatest negative effects on fecal output and also the greatest positive effects on mucus excretions in August. Our findings may be useful for suggesting feeding and housing strategies to enhance the welfare of captive pandas.

Documenting the causes of mucous stool occurrence in captive pandas has attracted remarkable attention and efforts from both scientists and zoo managers in the past. A previous study showed that mucous excretion in captive giant pandas was a husbandry- or management-related issue [6]. The *Technical Regulation of Husbandry and Management of the Giant Panda* (LY/T 2015-2012) requires that the minimum daily amount of bamboo provided to captive pandas should be 80% of their daily diet in subadult and adult pandas, and the percentage of crude fiber in their daily ration should not be less than 44.90% [19]. However, the amount of daily bamboo intake of the studied adult pandas at the Beijing Zoo was approximately 4.0 kg, which was only one-third of their conspecifics at the CCRCGP, and one-sixth of their wild conspecifics [20]. A higher occurrence of mucous stools has been observed in the pandas at the Beijing Zoo than that at the CCRCGP [21]. Thus, it could be speculated that the reduced daily fecal amounts of our pandas was directly caused by the low quantity of fiber in their dietary ration, which could have also led to the formation of mucous stools, as previously suggested in captive pandas [6,7]. Our findings are in line with the literature, confirming that there was a negative correlation between daily fecal output and the occurrence of mucous stools in captive pandas [5,22]. Clearly, it is also worth mentioning that a reduction in fecal output is expected in an animal that refuses to eat and is in pain, showing clinical signs of a colic syndrome. Consequently, to understand the underlying causative mechanisms of the excretion of mucous stools, further tests and data collection are needed.

The occurrence of mucous stools has also been proposed to be associated with the animals’ gender and age [11]. In the current study, however, we did not find a significant correlation between the occurrence of mucous stools and panda gender, but we confirmed the age effect. We found that elderly pandas usually excrete less feces each day and were more likely to suffer from excretion of mucous stools. This is highly consistent with a previous result by Liu XZ et al. [11]. 

The monthly effect on the mean frequency of the occurrence of mucous stools is a new finding. Both the peak of the mean frequency of mucus occurrence and the time factor analysis showed that the monthly effect may play a role in mucus excretion. However, the pattern of two periods is conflicting. This may be due to a mixed impact of weather and seasonal variation of bamboo. August is the hottest month in Beijing, and the high ambient temperatures impair the nutritional composition and palatability of bamboo [23]. Thus, there may have been a decreased bamboo intake in August for the pandas in this study, which could be the reason why August was positively correlated with mucus occurrence and negatively correlated with fecal output. The mucus occurrence peak in October might possibly be related to the diet transition from culm to leaves due to the feeding preferences of the pandas in this study, which may cause an increased occurrence of mucus excretion [4,5]. In addition, a feeding behavior study showed that the leaf was the primary plant part consumed from June to December, whereas culm was consumed primarily from February to May [24]. Another nutrition experiment showed that the total fiber was always greater in the culm than in the leaves throughout the year [25]. The reduction in fiber intake might explain the increasing trend in the frequency of mucus stools in October, even though the food intake remained the same. Nonetheless, future studies on the underlying causes for the synchronous and consistent changes in the mean fecal output and mean frequency of mucous stools in October are needed.

As of November 2018, the captive population of giant pandas is 548, and these pandas are raised in 93 institutions around the world [9]. These pandas mainly belong to the CCRCGP (approximately 53.08%) and the Chengdu Base of Panda Breeding (approximately 36.3%), which are both located in Sichuan Province, which has a rich supply of fresh bamboo resources. There are approximately 62 pandas, which account for 10.97% of the whole captive population, that are loaned or kept in institutions outside Sichuan. The native bamboo or favorite bamboo species that pandas like to eat will not be as available for pandas on loan, as they are grown within the historical home range of the pandas. Therefore, it is vital to consider how to increase the ratio of bamboo in their daily diets in order to keep the pandas healthy. *Technical Regulation of Husbandry and Management of the Giant Panda* (LY/T 2015-2012) should be strengthened for the care of captive, elderly pandas. It should be recommended that no less than 80% of the diet provided to captive pandas consists of bamboo, or the crude fiber content of the diet should be increased to no less than 45% by adding bamboo powder [21]. From the perspective of the welfare of elderly pandas, the daily fiber intake warrants special attention. Offering an adequate high-fiber diet may be a feasible method to address the increased frequency of mucous excretion as pandas age.

Our findings need to be interpreted with caution because this study had some limitations. The most important limitation was that we could not systematically monitor and evaluate the pure bamboo intake of each panda throughout the entire study. The real bamboo intake instead of the amount of bamboo provided is a key factor that may be related to the amount of feces elimination. Another limitation was that we could not weigh the mucus but only recorded the occurrences of the mucous stool excretions for each panda during the data collection. There may be some individual differences in the volume of the mucus stool excretions of different pandas, and the degree of discomfort for each panda may be closely related to the amount of mucus and the length of the period of the mucus excretions. Finally, we did not evaluate the microbiome and the behavior of the studied pandas. Consequently, our findings need to be ascertained in future investigations that should pay attention to the mechanisms of mucus excretions in captive pandas from both behavioral and microbiological viewpoints, evaluating their actual fiber intake.

## 5. Conclusions

Overall, the occurrence of mucous stools was negatively correlated with the output of feces excreted each day in captive giant pandas, as expected. The elderly pandas showed a higher occurrence of mucous stools excretion, which was also in line with the literature. The greatest influence of the monthly factor occurred in August, which might be related to the change in bamboo intake. There was one peak in the mean frequency of mucous stools in October, which might be related to a reduction in total fiber due to the diet transition from bamboo culm to leaves. Further studies are needed to address the mechanisms of mucous stools excretion by investigating microbiome changes in addition to systematically quantifying the daily bamboo intake. 

## Figures and Tables

**Figure 1 animals-09-00264-f001:**
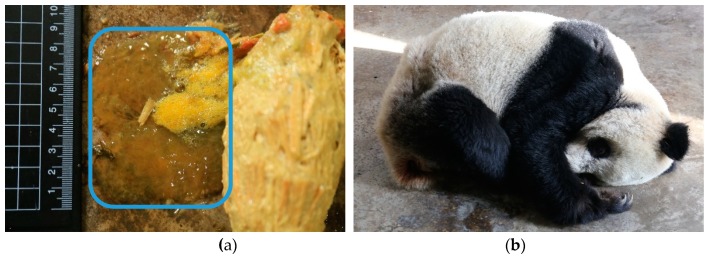
The mucus stool (**a**) and the uncomfortable panda (**b**) that recently finished mucus excretion (photo by Rongping Wei).

**Figure 2 animals-09-00264-f002:**
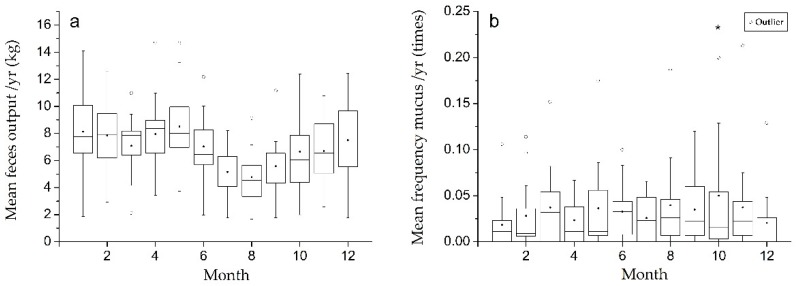
Average monthly changes in feces output (**a**) and mucus stool occurrence (**b**) for each month during the 15-year study. * indicates a significant peak.

**Table 1 animals-09-00264-t001:** Generalized estimating equations (GEE) model information (Algorithm converged).

Correlation Structure	Exchangeable
Subject Effect	Name (18 levels)
Number of Clusters	18
Correlation Matrix Dimension	5167
Maximum Cluster Size	5167
Minimum Cluster Size	280

**Table 2 animals-09-00264-t002:** Associations between excretion of mucous stools and age, gender, and feces output in captive giant pandas (*n* = 18) during a 15-year prospective study in China.

Parameter	Estimate	Standard Error	95% Confidence Limits		Odds Ratio	Z	*p* Value
**Intercept**	−3.2947	0.3718	−4.0233	−2.5660	-	−8.86	<0.0001
Gender	−0.6151	0.8157	−2.2138	0.9835	0.3509	−0.75	0.4508
Age	0.0589	0.0147	−0.0301	0.0876	0.5147	4.02	<0.0001
Feces	−0.1134	0.0275	−0.1674	−0.0594	0.4717	−4.12	<0.0001

**Table 3 animals-09-00264-t003:** Results of monthly changes of seasonal factors for mucus occurrences and fecal output in time series analyses.

Month	Mucus Occurrence	Fecal Output
January	−0.0075	1.475
February	−0.0037	0.975
March	0.0031	−0.106
April	−0.0022	0.341
May	0.0044	1.214
June	−0.0082	0.325
July	0.0068	−1.673
August	0.0126 *	−2.261 *
September	−0.0012	−1.215
October	0.0068	0.066
November	0.0074	0.067
December	−0.0048	0.792

* represents the biggest result by absolute value of seasonal factors by time series analysis.

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
