# Peer review of "Prospective Study on the Excretion of Mucous Stools and its Association with Age, Gender, and Feces Output in Captive Giant Pandas"

_animals, 2019, doi:10.3390/ani9050264_

Round 1
Reviewer 1 Report
This revised manuscript is improved but I still think there are significant problems with the analyses and some of the content. Specific comments below:
Delete lines 65-68. Unnecessary. The arguments made here and presented in the discussion that mucous stools can negatively impact the breeding or reintroduction program are very speculative. The authors do not present or cite any data suggesting that these events interfere with reproduction, and it seems that if the behavioral assay for reintroduction happens during the mucous event, it could skew results but given what we know about pandas, wouldn't one just do the assay at another time? These arguments must be more carefully worded, e.g. IF mucous events coincide with the single estrus that a female has each year on a regular basis, this COULD lead to negative effects on captive reproduction.
Lines 93-94. these lines cite some very old data; is it still true that captive pandas in china are not fed enough bamboo?
Line 141. Choosing July is entirely arbitrary and can lead to random results. If you picked another month, you'd get different months being significantly high or low, so these results are not at all reliable and can't be compared to monthly fecal output values. Instead take all of the monthly mucous values and calculate the mean, then look for any values that are more than 2 standard deviations from the mean. Remove those values and repeat the process until you have no more outlier values and then report which values were 2 SDs from the mean. See Brown et al 1994 in Zoo Biology 13:107-117 for a description of this technique, which is common approach to identifying peak values in hormone studies. Without changing the analyses of the data for Figure 2, it is not useful or valid. You might try to just run a simple correlation between monthly fecal out put and monthly mucous occurrence to see this they are correlated. This would corroborate your daily findings (or it might not). As was true in the previous version of this manuscript, the way the data in figure 2 are presented now, the results contradict your daily results somewhat because you have one "peak" mucous month during a time of high feces output (march) and one during a lower period (October) but still March and October are definitely NOT the lowest periods of fecal output. So without a formal correlational analysis at the monthly level of analysis, these conclusions are unfounded.
Table 3 is only referenced once and at the same time as Table 2 (on line162) so I wonder how much of table 3 is actually important to present. Still it is not clear what the labels under "factors" (e.g. OR Age) even mean. Tables in manuscripts have to make sense - they don't have to look just like a statistical program generates them. I would suggest deleting table 3 or merging some of the information in it with a revised Table 2
Line 171 and 173. You start with "it is shown" but then end with "data not shown" which is contradictory...
line 221, delete "which has not bee reported previously" because that is redundant with "new"
line 222 I don't think you can say we can understand the elevation in March because your monthly analyses do not actually correlate the two values and your method of picking which monthly values are high is not scientifically or statistically sound.
Line 239-251 As mentioned before, this is very speculative and needs to be stated more cautiously
Line 253. Earlier you said the technical manual already requires 80% bamboo, so how is your recommendation a "strengthening" of the manual?
Author Response
This revised manuscript is improved but I still think there are significant problems with the analyses and some of the content. Specific comments below:
Reply: Thanks lot for your comments and suggestions. Here’s our replies and answers. We’ve improved the deficiencies. Hoping this article is more qualified this time.
Delete lines 65-68. Unnecessary. The arguments made here and presented in the discussion that mucous stools can negatively impact the breeding or reintroduction program are very speculative. The authors do not present or cite any data suggesting that these events interfere with reproduction, and it seems that if the behavioral assay for reintroduction happens during the mucous event, it could skew results but given what we know about pandas, wouldn't one just do the assay at another time? These arguments must be more carefully worded, e.g. IF mucous events coincide with the single estrus that a female has each year on a regular basis, this COULD lead to negative effects on captive reproduction.
Reply: Done.Thank you for the advice. The unnecessary part has been simplified. Then we reorganized those speculative sentences in a cautious way.
Lines 93-94. these lines cite some very old data; is it still true that captive pandas in China are not fed enough bamboo?
Reply: Yes. Although the data is old, and the situation has improved, but the captive pandas are not fed enough fresh bamboo. According to reference 18, the foraging/feeding time of captive pandas is apparently shorter than their wild relatives. And as line 113, we also found pandas don’t eat enough bamboo (about 40%-50% of the provided). Moreover, there’s no recently published academic article reports the weight of the eaten bamboo, so we cited the old data here.
Line 141. Choosing July is entirely arbitrary and can lead to random results. If you picked another month, you'd get different months being significantly high or low, so these results are not at all reliable and can't be compared to monthly fecal output values. Instead take all of the monthly mucous values and calculate the mean, then look for any values that are more than 2 standard deviations from the mean. Remove those values and repeat the process until you have no more outlier values and then report which values were 2 SDs from the mean. See Brown et al 1994 in Zoo Biology 13:107-117 for a description of this technique, which is common approach to identifying peak values in hormone studies. Without changing the analyses of the data for Figure 2, it is not useful or valid. You might try to just run a simple correlation between monthly fecal out put and monthly mucous occurrence to see this they are correlated. This would corroborate your daily findings (or it might not). As was true in the previous version of this manuscript, the way the data in figure 2 are presented now, the results contradict your daily results somewhat because you have one "peak" mucous month during a time of high feces output (march) and one during a lower period (October) but still March and October are definitely NOT the lowest periods of fecal output. So without a formal correlational analysis at the monthly level of analysis, these conclusions are unfounded.
Reply: Agreed. The selection of July could have been arbitrary and may bias interpretation. We have taken your advice and deleted the Possion model, and used the 2-SD method to identify peaks in fecal output and mucus occurrences. In addition, we also have taken the second reviewer’s suggestion and performed a time-series analysis.
Table 3 is only referenced once and at the same time as Table 2 (on line162) so I wonder how much of table 3 is actually important to present. Still it is not clear what the labels under "factors" (e.g. OR Age) even mean. Tables in manuscripts have to make sense - they don't have to look just like a statistical program generates them. I would suggest deleting table 3 or merging some of the information in it with a revised Table 2
Reply: Done. Appreciate to your suggestion! We have combined table 2 and 3 into one table in the revised manuscript.
Line 171 and 173. You start with "it is shown" but then end with "data not shown" which is contradictory...
Reply: Agreed. We have amended the sentence that now it reads as follow: We also found that 15/17 individuals defecating in August significantly less than the annual median for the whole population at Beijing zoo throughout the 15 years study (data not shown).
“Data not shown” means we didn’t show all 17 statistics output of those pandas in the manuscript.
line 221, delete "which has not been reported previously" because that is redundant with "new"
Reply: Done. Thank you for putting forward this and we deleted the wordiness part.
line 222 I don't think you can say we can understand the elevation in March because your monthly analyses do not actually correlate the two values and your method of picking which monthly values are high is not scientifically or statistically sound.
Reply: This sentence has been amended accordingly with the new results generated with the new statistical analyses run.
Line 239-251 As mentioned before, this is very speculative and needs to be stated more cautiously
Reply: Thank you for pointing out those inexact sentences. We’ve replaced those speculative sentences.
Line 253. Earlier you said the technical manual already requires 80% bamboo, so how is your recommendation a "strengthening" of the manual?
Reply: As we previously explained, pandas won’t eat all the bamboo provided. In practice, enough bamboo will be provided to pandas, but they prefer to waiting for concentrated food. That’s why we said “strengthening”.

Reviewer 2 Report
The authors have undertaken to address the issues identified in the
previous review. However, the results are difficult to interpret and the
authors tend to overreach in finding significance for them. Although
they have somewhat tempered their conclusions, the article would likely
benefit from presenting the results as largely negative results - i.e.
no clear-cut useful relations can be established other than the higher
prevalence among older pandas, and the expectable reduced fecal output
on the days of mucous excretion.
The authors do not clarify why
or whether their unit of study are days with mucous excretion or events
of mucous excretion (which may span several days). Do mucous excretion
event always last the same in this study?
Figure 2 is central to presenting the results of this article: it would be easier to interpret if mucous excretion were presented as number of instances (or days?) of mucous excretion / month.
Figure two shows obvious asynchronous troughs of muchous excretion (dec-jan) and fecal output (july-sept). The latter seems linked to high ambient temperatures and decreased food intake. Might there also be an environmental cause for the trough in mucous excretion?
A boxplot would perhaps be more informative for figure 2.
The selection of July for statistical comparison seems arbitrary and may bias interpretation. Please reappraise the statistical analysis.
The multiyear analysis is confounded by the changes in individual animals. Performing a time-series analysis of months (with aggregated years) wouldbe more useful.
It might be worthwhile to explore analysis
of fecal output in the days preceding a mucous excretion (i.e. is
it normal / increased / declining?)
The manuscript needs to be revised to address these items, both in general and in specific parts, such as:
line 20: why is it expected?
line 160: the amount of feces DOES NOT INFLUENCE; there is an _inverse relation_ between amount of fecal output and occurence of mucous excretion. This is to be expected.
The manuscript also requires revision of English, as well as minor modifications, I have provided some examples below.
line 16: "uncomfortable and inactive state": poor English (the state is not uncomfortable, the panda is) -> discomfort and decreased activity acitivity.
line 20: unusual use of "phenomena", instances / cases / occurrences would be better
lines 28-30: very poor sentence structure
lines 30-31: verb-subject disagreement
lines 33-35: poor structure and poor English: perhaps change to this? "Eighteen giant pandas were studied at Beijing Zoo from April 2003 to June 2017 resulting in a total of 32856 daily fecal output records, and including 900 occurrences of mucous excretion."
line 38: is it typical because it is already reported - this is not mentioned.
line 135: include SAS "[...] of Proc Genmod (SAS) [...]"
throughout: I am unsure the usage of the possessive "pandas' " is correct, please check.
Author Response
The authors have undertaken to address the issues identified in the previous review. However, the results are difficult to interpret and the authors tend to overreach in finding significance for them. Although they have somewhat tempered their conclusions, the article would likely benefit from presenting the results as largely negative results - i.e. no clear-cut useful relations can be established other than the higher prevalence among older pandas, and the expectable reduced fecal output on the days of mucous excretion.
Reply: Thanks a lot for your comments and suggestions. We have amended the statistical analysis and we have tried to improve the clarity of the results section. We have also tried to interpreter our results in the discussion, damping down our conclusions.
The authors do not clarify why or whether their unit of study are days with mucous excretion or events of mucous excretion (which may span several days). Do mucous excretion event always last the same in this study?
Reply: The unit was days. It was an ambiguous expression and we clarified the expression of the mucous excretion in the manuscript (line 130). It was days. Thank you for putting forward this inexact expression.
Figure 2 is central to presenting the results of this article: it would be easier to interpret if mucous excretion were presented as number of instances (or days?) of mucous excretion / month.
Reply: Figure 2 has now be replaced based on the new statistical analysis.
Figure two shows obvious asynchronous troughs of muchous excretion (dec-jan) and fecal output (july-sept). The latter seems linked to high ambient temperatures and decreased food intake. Might there also be an environmental cause for the trough in mucous excretion?
Reply: There might be environment cause but we don’t have enough evidence and data to prove it. So we discussed about this and speculated it might be one of the causes.
A boxplot would perhaps be more informative for figure 2.
The selection of July for statistical comparison seems arbitrary and may bias interpretation. Please reappraise the statistical analysis.
The multiyear analysis is confounded by the changes in individual animals. Performing a time-series analysis of months (with aggregated years) would be more useful.
It might be worthwhile to explore analysis of fecal output in the days preceding a mucous excretion (i.e. is it normal / increased / declining?)
Reply: Agreed. Choosing July was arbitrary and may have lead to random result. So we have amended our statistical analysis. We deleted the Possion model then tried to perform a time-series analysis and used 2SDs to find peaks by following reviewer one’s suggestion. We have expressed the data as box plots.
The manuscript needs to be revised to address these items, both in general and in specific parts, such as:
line 20: why is it expected?
line 160: the amount of feces DOES NOT INFLUENCE; there is an _inverse relation_ between amount of fecal output and occurence of mucous excretion. This is to be expected.
Reply: The “expected” has been deleted. According to the author guidelines, we thought the simple summary targets to the public, so the language isn’t precise enough. Thank you for pointing out this mistake.
The manuscript also requires revision of English, as well as minor modifications, I have provided some examples below.
line 16: "uncomfortable and inactive state": poor English (the state is not uncomfortable, the panda is) -> discomfort and decreased activity acitivity.
line 20: unusual use of "phenomena", instances / cases / occurrences would be better
lines 28-30: very poor sentence structure
lines 30-31: verb-subject disagreement
lines 33-35: poor structure and poor English: perhaps change to this? "Eighteen giant pandas were studied at Beijing Zoo from April 2003 to June 2017 resulting in a total of 32856 daily fecal output records, and including 900 occurrences of mucous excretion."
line 38: is it typical because it is already reported - this is not mentioned.
line 135: include SAS "[...] of Proc Genmod (SAS) [...]"
Reply: We improved the content and structure and corrected the mistakes. Appreciate to your suggestions!
throughout: I am unsure the usage of the possessive "pandas' " is correct, please check.
Reply: We double checked and it is grammatically correct and it has been often used in other published stories and reports.

Round 2
Reviewer 1 Report
It is still curious why you get different results from the time series and monthly trend analyses. The biological explanations you offer to make sense in terms of biology but it is still unclear why the statistics come out differently. That said, there is actually some general agreement between two analyses: the time series analysis says August is the month with the lowest amount of fecal output likelihood and highest likelihood of mucous stools. Figure 2 does show that August is the lowest month in median fecal output but is maybe not significantly lower using the Brown method because of variability. But I also note that in line 146 you say you calculated standard errors, but that is not the Brown method. You would calculate the MEAN and STANDARD DEVIATION of all data and then see which months fall outside the mean +/- 2 SDs. So maybe that is why it didn't come out significant. Re-check calculations. August is also one of the higher months for mean occurrence of mucous stools (even though not significantly different) so that does agree with the time series analyses. I might point out that there are broad similarities between the two analyses after you re-check the analysis.
Author Response
It is still curious why you get different results from the time series and monthly trend analyses. The biological explanations you offer to make sense in terms of biology but it is still unclear why the statistics come out differently. That said, there is actually some general agreement between two analyses: the time series analysis says August is the month with the lowest amount of fecal output likelihood and highest likelihood of mucous stools. Figure 2 does show that August is the lowest month in median fecal output but is maybe not significantly lower using the Brown method because of variability.
Reply: Thank you for your comments and suggestions. It seems that you may misunderstand the result of the time-series analysis. The result showed the effect of time (change of month) instead of the change of amount (of feces/mucous changes over time) on the results of predicting the occurrence of mucous. The changes of feces/mucous amount are influenced by diet variation (such as nutrient), management and other factors. This may be the reason why the results of two statistics did not match completely.
But I also note that in line 146 you say you calculated standard errors, but that is not the Brown method. You would calculate the MEAN and STANDARD DEVIATION of all data and then see which months fall outside the mean +/- 2 SDs. So maybe that is why it didn't come out significant. Re-check calculations. August is also one of the higher months for mean occurrence of mucous stools (even though not significantly different) so that does agree with the time series analyses. I might point out that there are broad similarities between the two analyses after you re-check the analysis.
Reply: Thank you for pointing this mistake out. We re-checked the statistics for this part and found that we used the wrong words in describing this method in the previous version of manuscript. We used the standard deviation (SD) in running this statistic.

Reviewer 2 Report
The authors have provided a thouroughly revised manuscript which warrants publication. However, it still warrants minor corrections such as the following:
lines 85-86: goblet cells are not unique to panda bears
line 108: the surface notation used does not appear adequate (at the very least, insert spaces and uncapitalize the X: 5.0 x 10.0 m
lines 160-161: poor english, re-write (e.g.: eeg was run with SAS, and all other statistical analyses were run with spss)
line 219: it is unclear where the authors obtain the figure for kg of bamboo ingested at Beijing Zoo (4.0 kg), as they have stated (lines 112-116) that this is more likely to be in excess of 10kg (at least 40% of 25 kg bamboo offered).
line 264: "needs paid special attention" is not correct, substitute with warrants special attention
Author Response
The authors have provided a thouroughly revised manuscript which warrants publication. However, it still warrants minor corrections such as the following:
lines 85-86: goblet cells are not unique to panda bears
line 108: the surface notation used does not appear adequate (at the very least, insert spaces and uncapitalize the X: 5.0 x 10.0 m
lines 160-161: poor english, re-write (e.g.: eeg was run with SAS, and all other statistical analyses were run with spss)
line 219: it is unclear where the authors obtain the figure for kg of bamboo ingested at Beijing Zoo (4.0 kg), as they have stated (lines 112-116) that this is more likely to be in excess of 10kg (at least 40% of 25 kg bamboo offered).
line 264: "needs paid special attention" is not correct, substitute with warrants special attention
Reply: Thank you for your comments and suggestions. We’ve improved the contents according to your advice.
About the mismatching of bamboo ingested, the number in line 112-116 is based on our observation (mostly by zoo keepers including Xuefeng Liu, Tao Ma etc at Beijing Zoo and Guiquan Zhang at CCRCGP), another number (4kg) in line 219 was from a literature [Wang, P.; Li, D. Husbandry of Giant Pandas; China Forestry Publishing House: Beijing, 2003; page 57.]

This manuscript is a resubmission of an earlier submission. The following is a list of the peer review reports and author responses from that submission.
Round 1
Reviewer 1 Report
This is a correlational study of fecal output and occurrence of mucous stools in captive giant pandas. The authors argue that there is a relationship between fecal output and occurrence of mucous stools, yet their own results are contradictory to one another. The methods and results must be clarified to determine whether their results make sense. In one regard it is not surprising to find a relationship between occurrence of mucous stool and fecal output because during "mucous events" pandas often eat less and do not defecate very much or at all. So reduced fecal output is more a symptom of a mucous event rather than a cause.
What is confusing in the methods is that data were collected on a daily basis for both fecal output and mucous occurrence but then the statistics were done on mean amount of feces produced (lines 136-137). In my experience, mucous events last typically 2-4 days so that leaves a lot of other time in the month for other factors to contribute to fecal output. In my experience, each mucous event culminates in one mucous stool. So it isn't clear why we should look for a relationship between monthly fecal output and occurrence of mucous stools. The PIs should just focus on fecal output in the days leading up to and after mucous stool passage. I think this is also why their results appear to contradict one another as described below.
Given that there appears to be significant individual variation in the variables measured, I would expect to see an ID or Individual variable included in the models but it isn't clear that it was included.
First in line 158-159, they say that mucous peaks occur in Feb-March and Oct-Nov, which overlaps with periods of peak feces output (Feb to June and Oct to Dec) according to lines 155-159. So based on these statements, one could conclude that the highest frequency of mucous events occurs AT THE SAME TIME as the highest fecal output. But in line 163 they say that the statistics show that the amount of feces excreted DAILY (but early in line 136 they say mean monthly amounts of feces were the data used for analyses so this doesn't makes sense) was negatively correlated with occurrence of mucous. So the PIs need to untangle these two statements and clarify why they are not contradictory (if in fact they are not) and clarify what units of measurement were used to make each of these claims.
Using Figure 2 a,b (line 168) the authors claim that there is a seasonal change in feces output but the deviations from median here are VERY small (less than 1% at most) and show high variability. And later in line 171-172 they show that in fact there is no seasonal change except for the month of August. This contradicts earlier statements in lines 155-159 that there are seasonal trends. In addition, the authors fail to mention in this set of analyses that there is no seasonal effect on mucous production (figure 2D). So again I don't find conclusive evidence that there is 1) a firm relationship between fecal output and mucous occurrence as measured by the PIs (though i believe there is a relationship over small time frames (e.g. 2-4 days) and there seems to be no strong seasonal trend in fecal output and if there is, it doesn't seem to be related to mucous occurrence.
In Table 3, I have no idea what the entries in the "label" column mean or refer to. The figure title for figure 2 is not very clear. The introduction is too long. In the first paragraph I would start with the idea expressed in line 51 that diets fed to captive animals can affect their welfare. The preceding lines are not very useful. Lines 64-71 are also not very relevant. The claim that mucous stools can negative affect captive breeding is not very sound. We now know how to breed captive pandas very well and many cubs are produced every year in the large breeding centers, despite the occurrence of mucous stools.
The heat maps all printed very small so I couldn't make them out clearly. I don't know that is very useful to include them for each animal. Perhaps one big heat map that includes data from all of the animals would be more instructive (and may help to support the claims made by the authors).
Author Response
Please look at the attached word file.

Reviewer 2 Report
This article attempts to address the phenology of "mucous excretion" in captive pandas (Ailuropoda menlanoleuca) at Beijing Zoo over a period of 15 years. Specifically, the authors present data on the amount of feces produced by the studied pandas in relation to the occurence or absence of "mucous excretion". The data collected for this study provide insights into the phenology of fecal production and episodes of "mucous excretion" by the pandas at Beijing Zoo. However, the article has several failings that make it unsuitable for publication in its current state:
"mucous excretion" in other carnivore species is a hallmark sign of colitis, which would be in accordance to the malaise observed in the pandas which excrete mucous. In instances of colitis, a decreased appetitie and therefore decreased production of normal feces is to be expected. The authors fail to explore this expectable link between the observation of a sign of colitis and decreased feces in their evaluation and analysis.
the authors conclude that decreases in the ingestion of fiber leads to mucous excretion (lines 200-205). This is an unwarranted conclusion, as they have presented no temporal analysis to support a decreased ingestion of bamboo with subsequent "mucous excretion". If what was meant in this paragraph is that standing low consumption of bamboo may be related to a high incience of colitis as signalled by fecal mucus in the pandas at Beijing Zoo, this needs to be stated more clearly. However, this would bear little relation to the data presented.
claims such as dehydration to account for reduced fecal production (lines 228-230) are easy to substantiate and should not be presented as unproven hypothesis.
the text needs to be revised both for content and structure, some examples of which are:
the leading sentence in line 46 is not adequate
paragraph 2.2.1 needs to be reogranized
amend unsubtstantiated claims such as in line 71
English usage througout the article needs to be revised (e.g. using byproducts instead of consequences -line 49, incorrect verb tenses -line 198)
The data in this study appear to show no temporal relation between putative colitis and fecal production in the pandas studied. Interestingly, the pandas at Beijing Zoo show a marked reduction in these episodes during December and January. The article should concentrate on these findings, attempt to put them into context based on concrete information, including occurence of colitis in other carnivores), phenology of this problem at other institutions, and including in pandas housed outside of PRC, if possible. The possible effects of seasonal changes in bamboo (Knott 2017,
10.1371/journal.pone.0177582) should also be addressed.
Author Response
Please look at the attached word file with format.
